# Fresh and Browned Lotus Root Extracts Promote Cholesterol Metabolism in FFA-Induced HepG2 Cells through Different Pathways

**DOI:** 10.3390/foods12091781

**Published:** 2023-04-25

**Authors:** Shuyuan Zhong, Jingfang Li, Meng Wei, Zeyuan Deng, Xiaoru Liu

**Affiliations:** State Key Laboratory of Food Science and Technology, Nanchang University, Nanchang 330047, China; shuyuanz2022@163.com (S.Z.);

**Keywords:** browned lotus root, cholesterol metabolism, bile acid synthesis, cholesterol synthesis

## Abstract

Browning of fresh-cut plants is mainly attributed to the enzymatic browning of phenolic compounds induced by polyphenol oxidase (PPO), producing browning products such as anthraquinones, flavanol oxides, and glycosides, which are usually considered to be non-toxic. Could browning bring any benefits on behalf of their bioactivity? Our previous study found that browned lotus root extracts (BLREs) could reduce the cholesterol level in obese mice as fresh lotus root extracts (FLREs) did. This study aimed to compare the mechanisms of FLRE and BLRE on cholesterol metabolism and verify whether the main component’s monomer regulates cholesterol metabolism like the extracts do through in vitro experiments. Extracts and monomeric compounds are applied to HepG2 cells induced by free fatty acids (FFA). Extracellular total cholesterol (TC) and triglyceride (TG) levels were also detected. In addition, RT-PCR and Western blot were used to observe cholesterol metabolism-related gene and protein expression. The in vitro results showed that BLRE and FLRE could reduce TC and TG levels in HepG2 cells. In addition, BLRE suppressed the synthesis of cholesterol. Meanwhile, FLRE promoted the synthesis of bile acid (BA) as well as the clearance and efflux of cholesterol. Furthermore, the main monomers of BLRE also decreased cholesterol synthesis, which is the same as BLRE. In addition, the main monomers of FLRE promoted the synthesis of BAs, similar to FLRE. BLRE and FLRE promote cholesterol metabolism by different pathways.

## 1. Introduction

Cardiovascular disease (CVD) is a type of disease involving the heart or blood vessels. Atherosclerosis (AS)-induced CVD is a major cause of death worldwide, and high cholesterol levels are strongly associated with cardiovascular disease [1,2]. The essence of cholesterol metabolism is a series of enzymatic reactions, and the activity of some key enzymes determines the rate and direction of its metabolism. The liver synthesizes approximately 70% of the cholesterol in the body, and cholesterol homeostasis is achieved mainly through biotransformation, clearance, efflux, and cholesterol synthesis. BAs are synthesized from cholesterol via classical or alternative pathways in the liver. The rate-limiting enzyme cholesterol 7α-hydroxylase (CYP7A1) and sterol 12α-hydroxylase (CYP8B1) initiate the classical pathway. At the same time, the alternative pathway is initiated by mitochondrial sterol 27-hydroxylase (CYP27A1) and oxysterol 7-α hydroxylase (CYP7B1) [3]. Then, Farnesoid X receptor (FXR) is activated by BAs. It binds with the retinoid X receptor (RXR) and inhibits transcriptional repression of the CYP7A1 [4]. In addition to this, FXR is highly responsive to BAs and increases Fibroblast Growth Factor 19 (FGF19) when chenodeoxycholic acid (CDCA) is administered to humans [5]. The deposition of low-density lipoprotein cholesterol (LDL-C) harms CVD and requires LDL receptor (LDLR)-mediated endocytosis to clear from the liver [6]. Meanwhile, ATP-binding cassette (ABC) transporter A1 (ABCA1), ABCG1, and scavenger receptor BI (SR-BI) mediate cholesterol efflux from macrophages [7]. The synthesis of cholesterol occurs mainly in the liver. Numerous studies have proven that sterol-response element-binding protein 2 (SREBP2) and the rate-limiting enzyme HMG-CoA reductase (HMGCR) were the key regulators of cholesterol synthesis [2,8].

Natural compounds can effectively promote cholesterol metabolism by regulating the abovementioned genes and proteins. Hyunsook Kim Paul et al. fed a high-fat diet of Chardonnay grape seed flour to male golden Syrian hamsters, then observed down-regulation of intestinal FXR and Fibroblast Growth Factor 15 accompanied with up-regulation of CYP7A1 in the liver [9]. Paul F. Lebeau et al. found caffeine could inhibit the expression of the proprotein convertase subtilisin/kexin type 9 (PCSK9) and promote the expression of LDLR to accelerate the clearance of LDL-C in HepG2 cells [10]. Liang Jing et al. also found that, after administration of naringin to hyper-cholesterol HepG2 cells, the expression of LDLR and CYP7A1 was increased, and thus, cholesterol levels were reduced [11]. Ya-Ru Wu et al. proved that liraglutide could alleviate lipid deposition in high-fat-diet mice and significantly increased ABCA1 protein expression in HepG2 cells treated with high glucose [12]. In addition, garlic extracts improved cholesterol metabolism in lipopolysaccharides-induced HepG2 cells by down-regulating SREBP2 and HMGCR expressions and up-regulating the expression of LDLR [13]. Thus, this article will examine cholesterol metabolism in terms of its biotransformation, clearance, efflux, and synthesis.

Lotus root (*Nelumbo nucifera* Gaertn.) is rich in polyphenols and flavonoids, and is popular in China where it acts as a characteristic aquatic economic crop. It has been reported to have various biological activities such as antioxidant [14], immunomodulatory, antiviral [15], and hypoglycemic effects [16]. In most studies, organic solvents, water, microwaves, or ultrasound were mainly used to extract the active ingredients of lotus root. It showed that ethanol extract of lotus root (ELR) could inhibit lipid accumulation in high-fat-diet rats [17]. Yumi Tsuruta et al. found that hepatic steatosis in obese diabetic db/db mice was significantly alleviated by lotus root polyphenol extract [18]. As lotus roots are susceptible to mechanical damage during storage and processing, browning has always been inevitable and undesirable. After browning occurs in lotus roots, their appearance, flavor, and nutrients (phenolics, proteins, sugars, etc.) are significantly reduced. Thus, browning is usually considered harmful. Numerous studies have shown that the leading cause of browning in lotus roots is the release of phenolic enzymes, such as PPO, peroxidase, and phenylalanine aminolase, from the tissues after harvest. Then, the air and polyphenolic compounds in the lotus roots combine with those enzymes, resulting in enzymatic browning. The darkening is due to the transformation of the polyphenols into anthraquinones induced by PPO, which are further oxidized to form polymers [19,20]. In addition, anthraquinone was reported to be non-toxic and biologically active [21]; flavanol oxides have also been proven to be absorbed by human intestinal epithelial cells [22]; and the glycosides produced from sugars and phenolic acids were also non-toxic and exhibited biological activity [23]. Most studies have focused on lotus root browning and its inhibition, but the biological activity of browning products is rarely studied. Others also focused on the methods of inhibiting the browning of lotus roots. However, whether browning products have beneficial biological activities for the human body is worth exploring. In addition to this, comparative studies seldom exist between fresh and brown lotus roots [24,25]. Hence, based on our previous discovery that browned lotus root reduced cholesterol, in this paper, we aim to compare the similarities and differences in the mechanisms of the lipid-lowering effects of substances extracted from lotus roots before and after browning by ultrasound-assisted extraction with 80% methanol.

At present, studies on extracts of lotus root mainly focus on lipid metabolism but seldom on cholesterol metabolism. The liver is the leading site of cholesterol metabolism, and HepG2 cells have properties similar to those of normal human hepatocytes, such as the ability to synthesize, remove, and efflux cholesterol autonomously [26]. In addition, HepG2 cells have the advantages of more stable phenotypic traits and are easy to obtain. HepG2 cells are representative, highly recognized, and widely used experimental cell models for studying cholesterol metabolism and lipid metabolism in the liver [27]. Therefore, this study aims to compare the mechanisms of extracts of lotus root before and after browning on cholesterol metabolism with HepG2 cells. Moreover, the main monomers of the two sections were selected for comparative verification.

## 2. Materials and Methods

### 2.1. Preparation of Lotus Root Extracts before and after Browning and Identification of the Extracts

#### 2.1.1. Browning of Fresh Lotus Roots

The fresh lotus roots were purchased from Honghu, Hubei, China; its variety is pink root. After washing, the fresh lotus root was cut into 3–5 mm slices. Then, a portion was taken out, freeze-dried, and stored at −20 °C, while the remaining were placed in a constant temperature and humidity incubator for 3 days. Finally, the temperature was set at 20 °C and the humidity at 90%. After browning, they were freeze-dried and then stored at −20 °C.

#### 2.1.2. Extraction of Substances from Fresh and Browned Lotus Roots

Extraction is based on the method found in the literature with minor modifications [28]. First, weigh a certain amount of fresh or browned dried lotus root in a beaker, add 80% methanol (the ratio of material to liquid is 1:20), and stir thoroughly for ultrasonic extraction for 1.5 h (ultrasonic settings: time 30 min, power 90%, temperature 30 °C); then, centrifuge at 4500 r/min for 10 min with a high-speed centrifuge to extract the supernatant and concentrate by evaporation at 30~40 °C with a rotary evaporator. Subsequently, the extract is collected in a dish for freeze-drying, and finally, the extract powder is obtained. The fractions of the FLRE and BLRE were stored at −20 °C.

### 2.2. Cell Culture and Treatment

The HepG2 human hepatocellular carcinoma cell line was purchased from Shanghai Gefan Biotechnology. Co., Ltd., Shanghai, China (National Collection of Authenticated Cell Cultures). Dulbecco’s Modified Eagle’s Medium (DMEM) supplemented with 10% FBS, 100 U/mL penicillin, and 100 µg/mL streptomycin was used to culture the HepG2 cells in an incubator at 37 °C and maintained in a humidified atmosphere containing 5% CO_2_. The culture medium was changed daily, and cells were subcultured upon reaching approximately 80–90% confluence. Oleic acid (OA) was dissolved in 0.1 M NaOH solution, saponified at 70 °C for 30 min, mixed with 4 mM fatty-acid-free bovine serum albumin (BSA), and heated at 50 °C for 1 h. Sodium palmitate was dissolved in sterile water, heated at 70 °C for 30 min, then mixed with 10% BSA, heated at 50 °C for 1 h, and finally mixed with OA (PA: OA = 2: 1) in DMEM to form 1 mM FFA. The prepared FLRE and BLRE were dissolved in sterile water and diluted in DMEM. The four selected monomers (catechin, caffeic acid, theaflavin, and forsythoside A, purchased from Solarbio, Beijing, China) were dissolved in DMSO and diluted in DMEM. When the cells reached about 50% confluence, they were treated with 1 mM FFA for 24 h. At the same time, cells were treated with a solvent mixture of palmitic acid and oleic acid at a ratio of 2:1 for 24 h as the control group. Then, they were treated with FLRE, BLRE, and monomeric compounds for another 24 h, while the control group was treated with sterile water for another 24 h.

### 2.3. Cell Viability Assay

HepG2 cells with 80–90% confluence were inoculated in 96-well plates (1 × 10^4^ cells/well) and treated with FFA (0, 0.4, 0.6, 0.8, 1, 1.2, and 1.4 mM) or FLRE and BLRE (0, 0.1 mg/mL, 0.2, 0.4, 0.6, 0.8, and 1 mg/mL) or compounds (0, 0.1, 0.2, 0.4, 0.6, 0.8, and 1 mM) diluted in DMEM for 24 h, respectively. Subsequently, cell viability was determined using the Cell Counting Kit-8 Assay (Beyotime Biotechnology Co., Ltd., Shanghai, China) according to the manufacturer’s instructions. After pouring off the culture medium in the 96-well plates and washing twice with PBS, 100 µL of 10% CCK-8 was added to each well and incubated at 37 °C for 30 min~1 h; then, the absorbance of each well was measured at 450 nm using MicroplateReader. Cell viability = [absorbance (experimental hole − blank hole)/(control hole − blank hole)] × 100%.

### 2.4. Measurement of Intracellular TC and TG Levels

HepG2 cells grown to 80–90% confluence were inoculated in 6-well plates (5 × 10^5^ cells/well). HepG2 cells were pre-treated with FFA and stimulated with the two extracts or the monomer compounds for 24 h. The cell precipitates were crushed in an ultrasonic cell crusher. Then, cells were assayed for TC and TG according to the total cholesterol and triglyceride kits (Nanjing Jiancheng Biotechnology Co., Ltd., Nanjing, Jiangsu, China). All procedures were carried out according to the instructions of the kits.

### 2.5. RT-PCR

Cells were inoculated in 12-well plates (2 × 10^5^ cells/well), pre-treated with FFA, and stimulated with the two extracts or the monomer compounds for 24 h, respectively. Total RNA was extracted by lysing the cells with Trizol reagent (ThermoFisher Scientific, Waltham, MA, USA). Total RNA concentration and purity were measured by NanoDrop 8000 micro-UV-Vis (ThermoFisher Scientific, MA, USA) spectrophotometer. The cDNA was reverse transcribed using a TECHNE TC512 gradient PCR instrument (Bio-techne, MN, USA) under the following reaction conditions: 2 min at 42 °C and 15 min at 37 °C. The fluorescent PCR reaction system goes for: 4.4 µL for cDNA, 5 µL for ChamQ Universal SYBR qPCR Master Mix, and 0.3 µL each for upstream and downstream primers (10 µmol/L), for a total of 10 µL of the mixed reaction system. Fluorescence amplification was carried out in a CFX96 Touch real-time quantitative PCR instrument (Bio-Rad, Hercules, CA, USA). The reaction conditions were as follows: 30 s reaction at 95 °C, 30 s at 60 °C, 40 cycles, 10 s at 95 °C, 30 s at 60 °C, and 10 s at 95 °C. The relative expression levels were calculated by the 2^−ΔΔCt^ method using glyceraldehyde-3-phosphate dehydrogenase (GAPDH) as an internal reference. The relevant primer sequences are shown in Table 1.

### 2.6. Western Blot

The cells were inoculated in 6 cm Petri dishes, pre-treated with FFA, and stimulated with the two extracts or the monomer compounds for 24 h. A 200 µL volume of cell lysate (RIPA: protease inhibitor: PMSF = 100:1:1) was added to each dish, and the lysate was prepared and ready to use. After lysis on ice for 30 min, the cells were removed to EP tubes. The supernatant was removed by centrifugation at 4 °C to obtain the protein. The protein concentration was then determined using the BCA Protein Concentration assay kit. Then, 1/5 volume of 6× loading buffer was added quantitatively and denatured in a boiling water bath for 5 min. The same amount of protein was separated from the sample by SDS-PAGE electrophoresis (sample volume 30–40 μg) (Bio-Rad, CA, USA), then transferred to a PVDF membrane activated with anhydrous methanol and cut according to the molecular weight of the target protein. The cut strips were closed at room temperature for 2 h on a shaker. The target strips were washed 3 times and then incubated with primary antibodies GAPDH (1:1000), NR1H4 (1:1000), FGF19 (1:1000), CYP7A1 (1:1000), SREBP2 (1:4000), and LDLR (1:1000) at 4 °C overnight. All antibodies were purchased from abcam Co., Ltd., Cambridge, UK and Wuhan East Lake Hi-tech Development Zone, Hubei, China. The target strips were washed 3 times with 1 × TBST, then incubated with secondary antibodies (mouse and rabbit antibodies) for 2 h at room temperature and washed with 1 × TBST. Then, the strips were blotted with filter paper and coated with ultrasensitive luminescent solution (liquid A:liquid B = 1:1) for 1–3 min, protected from light. The target protein was then analyzed using Image Lab 5.2 software.

### 2.7. Statistical Analysis

SPSS 20.0 software (IBM Corp. Armonk, NY, USA) was used for statistical analysis. Results were expressed as mean ± standard error of measurement (SEM). Differences between groups were analyzed by one-way ANOVA. *p <* 0.05 was considered to be statistically significant.

## 3. Results and Discussion

### 3.1. Selection of Monomer Compounds in FLRE and BLRE

According to the preliminary work of our research group, FLRE mainly contained catechin and caffeic acid. BLRE mainly contained theaflavin and forsythoside A. In terms of its structure, theaflavin is composed of a hydroxyl-substituted benzo ketone ring, which is formed by the condensation of the hydroxylated B-ring of catechin with the benzo ring of another catechin. Catechin belongs to flavanols, is the main component of tea polyphenols, and the basic carbon frame of its chemical structure is C6-C3-C6. In addition, it has a hydroxyl group at carbon 5 and 7 on the A-ring and has an ortho-dihydroxyl group in the B-ring at carbons 3′ and 4′ and a hydroxyl group at carbon 3 on the C-ring [29]. In addition, it has been reported that theaflavin in black tea results from the catalysis of catechin induced by polyphenol oxidase leading to its polymerization [29,30,31,32]. Moreover, the forsythoside A is synthesized by the condensation reaction of sugar, caffeic acid, and hydroxytyrosol [33]. Thus, the forsythoside A is derived from the caffeic acid. A study on the degradation mechanism of forsythoside A proved it could cyclize at high temperatures, producing suspensaside A after losing two hydrogen atoms; subsequently, rhamnosyl, phenyl ethanol, and glucosyl were removed; and, finally, caffeic acid is produced [34]. Finally, we selected catechin and caffeic acid to verify the effect of FLRE and theaflavin and forsythoside A to verify the effect of BLRE.

### 3.2. Cell Viability Assay to Determine the Doses of Each Group

As shown in Figure 1a, free fatty acids at concentrations of 0 to 1.4 mM had no significant effect on HepG2 cell viability and did not affect normal cell growth and proliferation. FLRE at concentrations of 0 to 0.6 mg/mL and BLRE at concentrations of 0 to 1 mg/mL have been observed not to affect the viability of HepG2 cells at 24 h (Figure 1b,c). The results in Figure 1b show that the viability of HepG2 cells was significantly inhibited when the concentration was 0.8–1 mg/mL. The viability of HepG2 treated with this concentration could still reach 89 ± 1% after 24 h. Although there was little effect on cell growth, the overall cell survival rate was higher. Thus, we selected three concentrations of 0.2, 0.6, and 1 mg/mL for the subsequent experiments of the two extracts. The concentrations of catechin ≤ 1 mM, caffeic acid ≤ 0.8 mM, theaflavin ≤ 0.2 mM, and forsythoside A ≤ 0.6 mM had no significant inhibition of cell proliferation (Figure 1d–g). Therefore, the four compounds’ concentrations of 0.2 mM were selected for follow-up testing according to their cytotoxicity.

In the present study, we modeled high cholesterol in HepG2 cells with a 1 mM concentration of FFA, which is generally used for in vitro fatty liver cell model establishment [35]. However, high concentrations of free fatty acids can easily cause cellular lipotoxicity and lead to apoptosis. On the other hand, it has been shown that 1 mM FFA significantly enhanced cellular lipid aggregation without affecting the average growth and proliferation of cells [36]. Therefore, 1 mM can be used as an appropriate concentration for HepG2 cells to establish a model of high cholesterol.

### 3.3. Effect on TC and TG

As can be seen from Figure 2a, the cholesterol content of the 1 mM FFA-treated group was significantly higher than that of the control group. Moreover, the total cholesterol content was significantly reduced after treating with different concentrations of FLRE and BLRE. Consistently, the lowest level of TC was observed in the extracts of the high-concentration group. As shown in Figure 2b, the TG content had been significantly decreased by 0.2 mg/mL FLRE and 0.2 and 1 mg/mL BLRE compared to the model group. To further clarify whether the main components of the extracts have the same lipid-lowering and cholesterol-lowering effects as the extracts, we investigated the impact of the monomer compounds on FFA-induced HepG2 cells. The results suggested that the treatment of catechin, theaflavin, and forsythoside A significantly decreased TC and TG content compared to the model group. Caffeic acid supplements could reduce intracellular TC content but had no significant effect on the content of TG (Figure 2c–f)).

Our results showed that the intracellular cholesterol and triglyceride levels were significantly reduced when pre-treated with FFA and then treated with FLRE and BLRE. At the same time, BLRE lowered triglycerides more significantly than FLRE. In addition to this, one study proved that fresh lotus root extracts could decrease the TC and TG levels in vivo [17]. Some studies have also shown that the polyphenolic extract of lotus root could reduce the TG content of the liver in obese mice [18]. Meanwhile, we also found that BLRE could improve FFA-induced lipid deposition in HepG2 cells and BLRE had a better lipid-lowering effect than FLRE. At the same time, the results of single compound verification experiments showed that, compared with the model group, the four groups of catechin, caffeic acid, theaflavins, and forsythin A significantly reduced TC levels. However, except for the caffeic acid group, the other catechin groups, theaflavin, and forsythoside A significantly reduced the TG level compared to the model group. A clinical study also suggested that, after 3 months of consuming catechin-rich beverages, these volunteers’ cholesterol levels were able to be decreased [37]; animal experiments also proved that caffeic acid and theaflavin supplementation could reduce the content of serum cholesterol in mice [38,39]; and an in vitro study suggested that theaflavins inhibited cholesterol incorporation in soluble micelles [40]. Our results were consistent with these previous reports. These results indicated that catechin and caffeic acid were the critical substances in FLRE that play a hypolipidemic role. In addition, theaflavin and forsythoside A are the vital substances in BLRE that play a role in lowering blood lipids.

### 3.4. FLRE Could Motivate the Synthesis of Bile Acid through FXR/FGF19-CYP7A1/CYP27A1 Pathway

#### 3.4.1. Effect of FLRE and BLRE on the Expression of Genes and Proteins Related to Cholesterol Biotransformation

To compare the similarities and differences in the mechanisms of cholesterol metabolism in lotus roots before and after browning, RT-PCR and Western blot were used to detect the expression of genes and proteins related to cholesterol biotransformation. As shown in Figure 3a,b, the model group exhibited up-regulated expression of genes in the synthesis of BAs (FXR, FGF19, CYP27A1, CYP7B1) and down-regulated classical pathways of BA metabolism (CYP7A1 and CYP8B1). Regarding protein expression, the model group significantly increased the expression of CYP7A1 and decreased the expression of FGF19. However, there was no significant difference in FXR expression compared with the control group (Figure 3c,d). As shown in Figure 3a,b, compared with the model group, all three concentrations of FLRE could reduce the expression of FXR and FGF19 mRNA and increase the expression of CYP27A1 and CYP7B1 mRNA. Interestingly, both 0.6 mg/mL and 1 mg/mL FLRE supplementation up-regulated the gene expression of CYP7A1. In addition, only 1 mg/mL FLRE significantly aggrandized the CYP8B1 mRNA expression. As shown in Figure 3c,d, the results showed that the protein expression of FXR and CYP7A1 is consistent with gene expression. The FGF19 protein expression was reduced by 0.6 and 1 mg/mL FLRE. The results of RT-PCR and Western blot assays indicated that FLRE might reduce cholesterol levels by promoting the conversion of cholesterol to bile acids. As shown in Figure 3a,b, compared to the model group, three concentrations of BLRE up-regulated FGF19 mRNA expression and down-regulated the gene expression on crucial enzymes of BAs synthesis (CYP7A1, CYP8B1, CYP27A1, and CYP7B1); the 0.2 and 1 mg/mL BLRE supplementation down-regulated FXR mRNA expression. In addition, it can be seen in Figure 3c,d that FXR protein expression was significantly down-regulated by the BLRE group. In contrast, FGF19 protein expression had no significant difference in the BLRE group. Moreover, 0.6 and 1 mg/mL BLRE could decrease the protein expression of CYP7A1.

#### 3.4.2. Effect of the Principal Monomer Components from FLRE and BLRE on the Expression of Genes and Proteins Related to Cholesterol Biotransformation

To verify whether the four monomer compounds corresponding to FLRE and BLRE have the same effect on cholesterol metabolism as the extracts, we examined the effects of catechin, caffeic acid, theaflavin, and forsythoside A on genes and proteins related to bile acid metabolism. As shown in Figure 4a,b, compared with the model group, the results show that FXR and FGF19 mRNA expression was down-regulated after catechin and caffeic acid treatment. Regarding the classical and alternative pathways of BA metabolism, the effects of the caffeic acid did not agree well with the extracts. Only catechin could increase the CYP7A1 and CYP7B1 mRNA expression. However, the Western blot results showed consistency in protein expression with the FLRE. Both catechin and caffeic acid administration down-regulated FXR and FGF19 and up-regulated CYP7A1 (Figure 4e,f). As shown in Figure 4c,d, compared to the model group, the compound groups (theaflavin and forsythoside A) reduced FXR mRNA expression without affecting the FGF19 mRNA expression. In addition, the compound groups reduced the gene expression of enzymes critical to BA synthesis (CYP7A1, CYP8B1, CYP27A1, and CYP7B1). As displayed in Figure 4g,h, the FXR and FGF19 protein expression trends of the theaflavin and forsythoside A were the same as BLRE. Nevertheless, the protein expression of CYP7A1 was decreased by theaflavin and forsythoside A.

The liver is the leading site of cholesterol metabolism. In the liver, FXR, FGF19, CYP7A1, CYP8B1, CYP27A1, CYP7B1, SREBP2, HMGCR, LDLR, and ABCA1 are genes that regulate cholesterol homeostasis and maintain a complex genetic network of cholesterol biotransformation, synthesis, clearance, and efflux [41,42,43]. When the bile acid level in hepatocytes is high, FXR is activated. At the same time, the downstream effect factor of FXR, fibroblast growth factor 15/19 (FGF15/19), is activated, which inhibits the expression of the CYP7A1 gene in the liver, thus reducing the synthesis of bile acid in the liver [44,45,46,47]. CYP7A1 and CYP8B1 as well as CYP27A1 and CYP7B1 were the rate-limiting enzymes in the classical and alternative pathways for synthesizing BAs in the liver, respectively. The classical pathway mainly synthesizes CA and CDCA, and the alternative pathway primarily produces CDCA [48]. Moreover, these rate-limiting enzymes were subject to FXR regulation [49]. In this study, we found that FLRE could down-regulate the expression of FXR and FGF19. Although the BLRE negatively regulated the expression of FXR, the FGF19 had increased. In addition to this, we found that the FLRE could positively regulate the expression of CYP7A1, CYP8B1, CYP27A1, and CYP7B1 while the BLRE could negatively regulate them. Therefore, FLRE could promote BA synthesis while the BLRE inhibited BA synthesis. Previous studies have found that pu’er tea maintains cholesterol homeostasis by inhibiting the intestinal FXR/FGF15 signaling pathway in obese mice and promoting the classical synthesis pathway of hepatic bile acids [41]. It has also been shown that hypercholesterolemia in TH mice may increase the amount of BA by inhibiting the feedback regulation of FXR/FGF15/FGFR4 signaling-mediated CYP7A1 [50]. Hence, we further inferred that FLRE induced the expression of CYP7A1 by targeting FGF19 and increased BA synthesis. However, further experiments are required to clarify the mechanisms of FLRE on FGF19. In this manuscript, we also found that the theaflavin and forsythoside A reduced the expression of FXR and had no remarkable effect on FGF19 mRNA expression. In addition, the theaflavin and forsythoside A had the same affected trends on CYP7A1, CYP8B1, CYP27A1, and CYP7B1 as BLRE did. The results indicated that BLRE and its compounds could not promote the synthesis of BAs. In addition, the results of the monomer compounds showed that the catechin and caffeic acid could decrease the protein expression of FXR and FGF19 as the FLRE did. Nevertheless, the catechin could positively regulate the expression of CYP7A1, CYP27A1, and CYP7B1. The caffeic acid also increased the expression of CYP7A1. An in vitro study using HepG2 cells also showed that catechins could up-regulate CYP7A1 mRNA levels [51]. A study on catechin compounds also found that EGCG reversed the FXR activation induced by chenodeoxycholic acid in vitro [52]. A recent study found that charcoal herb extract complex (CHC), largely represented by phenolic acids (caffeic acid and vanillin), significantly increased the expression of CYP7A1 and PXR (it can be combined with FXR) in primary broiler hepatocytes [53]. Thus, these results indicated that catechin and caffeic acid could promote BA metabolism, including FLRE.

### 3.5. FLRE Could Promote the Clearance and Efflux of Cholesterol by Activating LDLR and ABCA1

#### 3.5.1. Effect of FLRE and BLRE on LDLR and ABCA1 Expressions

To further elucidate the mechanisms underlying the changes in cholesterol levels in HepG2 cells, we also examined the expression of genes and proteins related to cholesterol efflux and clearance of LDL-C. As shown in Figure 5a, the gene expression of LDLR was inhibited by the FFA, but this inhibition was substantially reversed by the treatment with 1 mg/mL FLRE. The expression of ABCA1 was obviously promoted in 0.2 and 1 mg/mL FLRE groups compared with the model group. As shown in Figure 5b, treatment with 0.6 and 1 mg/mL FLRE was able to up-regulate the protein levels of LDLR. In addition, we also evaluated the effect of BLRE on FFA-induced cholesterol clearance and efflux in HepG2 cells. Different from FLRE, the results showed that, compared with the model group, 0.6 and 1 mg/mL BLRE could significantly reduce the gene expression of LDLR, and 1 mg/mL BLRE could reduce the gene expression of ABCA1 (Figure 5a). However, BLRE did not affect LDLR protein expression.

#### 3.5.2. Effect of the Principal Monomer Components of FLRE and BLRE on LDLR and ABCA1 Expression

We examined the gene expression of LDLR and ABCA1 of four monomer compounds. As shown in Figure 6a, gene expression of LDLR and ABCA1 had no effects after treating with catechin and reversed effects after treating with caffeic acid. These results are not in keeping with the FLRE. The gene expression of LDLR and ABCA1 was significantly down-regulated after theaflavin and forsythoside A treatment compared to the model group (Figure 6b).

It is well known that SREBP2 positively regulates LDLR gene expression, and increased expression of LDLR is followed by increased uptake of food-derived or endogenous LDL, thereby reducing plasma levels of LDL-C [54]. The proprotein convertase subtilysin 9 (PCSK9) can bind directly to the LDLR structural domain, resulting in a conformational change in the LDLR protein, which cannot return to the cell membrane surface and enters the lysosome directly for degradation [55]. The role of ABCA1 in hepatocytes is to allow the hepatic cholesterol to enter the bloodstream in the form of HDL-C. At the same time, activation of FXR also reduces ABCA1 and plasma HDL-C levels in the liver [56]. This study showed that the FLRE both up-regulated the expression of LDLR and ABCA1. The BLRE down-regulated LDLR and ABCA1 gene expression while having no significant effect on LDLR protein expression. It has also been reported that the active substance 6-gingerol in ginger mediates cholesterol metabolism through two pathways: firstly, by activating SREBP2 to up-regulate the gene and protein expression of LDLR to reduce cholesterol levels; secondly, by up-regulating the cholesterol-efflux-related genes LXRα and ABCA1 to reduce cholesterol levels [57]. Studies have found that Chinese olive (*Canarium album* L.) fruit extract, rich in gallic acid and ellagic acid, could simultaneously up-regulate the expression of LDLR and ABCA1 and promote cholesterol metabolism [58]. These results suggested that the FLRE also accelerated LDL-C clearance and cholesterol efflux. However, the catechin and caffeic acid did not increase LDLR and ABCA1 expression, which did not align with the FLRE. Meanwhile, the theaflavin and forsythoside A down-regulated LDLR and ABCA1 gene expression. A study about green tea polyphenols, which are rich in various catechins, proved that treatment with GTPs (green tea polyphenols) that contain ester bonds could increase the expression of LDLR. In contrast, GTP without ester bonds did not sufficiently induce LDLR accumulation [59]. An in vitro study also found the binding activity of catechins towards LDL-C with the ability in the sequence of (−)-epicatechin gallate > (−)-catechin gallate > (−)-epigallocatechin gallate while (+)-catechin, (−)-epicatechin, (−)-epigallocatechin, and (−)-gallocatechin showed low binding activity [60]. A systematic study on coffee also found that caffeic acid could increase the mRNA and protein levels of ABCG1, while the expression of ABCA1 could not be influenced [61]. Thus, we inferred that it was not the catechin and caffeic acid but other compounds in FLRE that promoted cholesterol clearance and efflux.

### 3.6. BLRE Could Regulate SREBP2-HMGCR-Mediated Synthesis of Cholesterol

#### 3.6.1. Effect of FLRE and BLRE on Cholesterol Synthesis Related Genes and Protein Expression

HMGCR plays a vital role in regulating the expression of genes related to cholesterol synthesis, while SREBP2 is upstream of HMGCR [62]. We examined whether lotus root extracts would modulate its expression in HepG2 cells. As shown in Figure 7a, treatment with FLRE caused a significant elevation in SREBP2 mRNA compared to the model group. Only 1 mg/mL FLRE supplementation enhanced the gene expression of HMGCR compared with the model group. As shown in the Western blotting analysis (Figure 7b), protein expression of SREBP2 was promoted by 1 mg/mL FLRE. Treatment with 0.6 and 1 mg/mL BLRE significantly inhibited SREBP2 mRNA and protein expression compared to the model group. In comparison, 0.2 mg/mL BLRE had no significant effect on SREBP2 mRNA and protein expression (Figure 7). As shown in Figure 7b, treatment with BLRE caused a significant reduction in HMGCR mRNA over-expression induced by FFA.

#### 3.6.2. Effect of the Principal Monomer Components from FLRE and BLRE on Cholesterol-Synthesis-Related Genes and Protein Expression

Validation experiments of the monomer compounds showed that catechin and caffeic acid did not affect gene expression of SREBP2. However, treatment with caffeic acid markedly reduced the expression of HMGCR compared with the model group (Figure 8a). Figure 8c shows that only catechin supplementation could significantly increase SREBP2 protein expression, while caffeic acid did not influence it. Furthermore, we also investigated the effects of theaflavin and forsythoside A on SREBP2 and HMGCR expression. As shown in Figure 8b,d,e, the data indicates that the gene and protein expression of SREBP2 was inhibited by the theaflavin and forsythoside A compared to the model group. Likewise, the gene expression of HMGCR was also down-regulated by theaflavin and forsythoside A compared to the model group.

SREBP2 is a critical transcription factor that regulates the synthesis and uptake of cholesterol. When intracellular cholesterol levels are low, SREBP2 is activated. Subsequently, it binds specifically to cholesterol-regulatory element-1 (SRE-1) on the HMGCR gene, promoting the expression of the HMGCR gene and resulting in increased cholesterol synthesis [63]. In our study, the expressions of SREBP2 and HMGCR were up-regulated by the FLRE. It is also reported that white tea extract, rich in catechins, could significantly enhance SREBP2 and LDLR expression, thereby decreasing cholesterol levels in HepG2 cells [64]. In comparison, SREBP2 and HMGCR expressions were down-regulated by the BLRE. A study indicated that transthyretin motivated cholesterol metabolism by inducing SREBP2-HMGCR and inhibiting LXRα-CYP7A1 [44]. It was reported that CCGQ (a mixture of saffron glucoside, chlorogenic acid, geniposide, and quercetin) treatment of HepG2 cells resulted in increased gene expression of ABCA1, CYP7A1, and AMPKα2, along with decreased expression of SREBP2, HMGCR, and LXRα genes, leading to a significant reduction in lipid deposition [65]. These results indicated that the BLRE could inhibit the SREBP2-HMGCR pathway to decrease cholesterol synthesis. In addition, the catechin supplement promoted the protein expression of SREBP2, which was consistent with FLRE. Monyuan Yang et al. also found that caffeic acid significantly enhanced the activation of AMPK and mediated a decrease in SREBP1, SREBP2, FAS, and HMGCR in OA-induced HepG2 cells [66]. Our study also found no effect of caffeic acid on SREBP2 but it down-regulated the HMGCR gene, which is inconsistent with FLRE. This further suggested that other cholesterol-lowering substances may be present in FLRE. In contrast, theaflavin and coniferin A significantly down-regulated SREBP2, HMGCR, and BLRE. As we all know, theaflavin is one essential bioactive compound of black tea [67]. Dev K. Singh et al. found that HMGCR reductase, when added to microsomal preparations, was directly inhibited by black tea extracts in McARH7777 rat hepatoma cells [68]. There are more studies on the inflammatory, oxidative stress, liver injury, and antibacterial activities of forsythoside A, and studies on cholesterol metabolism have not been reported [69]. Moreover, we found that the theaflavin and forsythoside A could suppress the synthesis of cholesterol, which is consistent with BLRE.

## 4. Conclusions

In conclusion, this study innovatively focused on the biological activity of plant enzymatic browning products. In vitro results revealed that the extracts of lotus roots after browning had a cholesterol-lowering effect just as the fresh lotus root extracts did, but with different metabolic mechanisms and pathways. Fresh lotus root extracts ameliorated cellular cholesterol elevation through at least two pathways: by inhibiting FXR/FGF19 signaling-pathway-mediated CYP7A1/CYP27A1 feedback regulation to promote cholesterol conversion to bile acids; and by activating SREBP2 to up-regulate LDLR, maintain cholesterol homeostasis, and promote LDL-C clearance and cholesterol-efflux-associated ABCA1 gene expression. The browned lotus root extracts mainly regulate cholesterol metabolism by inhibiting the expression of SREBP2 and HMGCR, genes related to cholesterol synthesis. In addition, catechin and caffeic acid are vital substances in fresh lotus root that promote cholesterol metabolism, and theaflavin and forsythoside A are crucial substances in browned lotus root.

Our research found that the browning product still has a hypolipidemic effect. These studies will help develop a natural source of compounds from lotus roots for treating chronic diseases associated with hypercholesterolemia and provide data to support research on browned lotus roots. At the same time, it will improve the utilization of browned lotus roots due to processing. However, more detailed and rigorous clinical studies are needed to verify their specific metabolic mechanisms.

## Figures and Tables

**Figure 1 foods-12-01781-f001:**
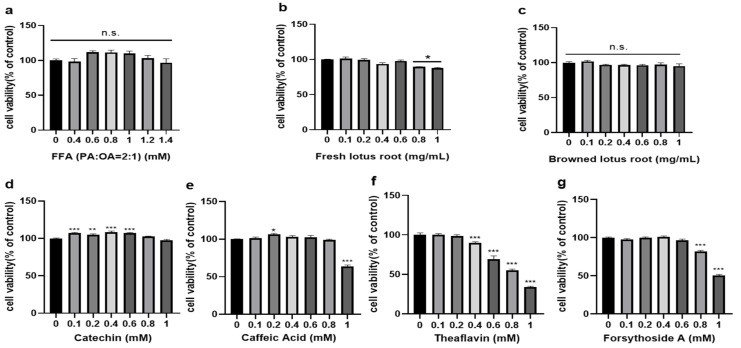
Effects of FFA (**a**), FLRE (**b**), BLRE (**c**), catechin (**d**), caffeic acid (**e**), theaflavin (**f**), and forsythoside A (**g**) on the viability of HepG2 cells. HepG2 cells were exposed to different concentrations of FFA (0, 0.4, 0.6, 0.8, 1, 1.2, 1.4 mM), FLRE and BLRE (0, 0.1, 0.2, 0.4, 0.6, 0.8, 1 mg/mL), and the monomer compounds (0, 0.1, 0.2, 0.4, 0.6, 0.8, 1 mM) for 24 h. After incubation for 24 h, cell viability was determined by CCK8 assay. Untreated cells served as the control. Compared with the control group, n.s., not significant. Significant at * *p* < 0.05, ** *p* < 0.01 and *** *p* < 0.001 compared with the control group.

**Figure 2 foods-12-01781-f002:**
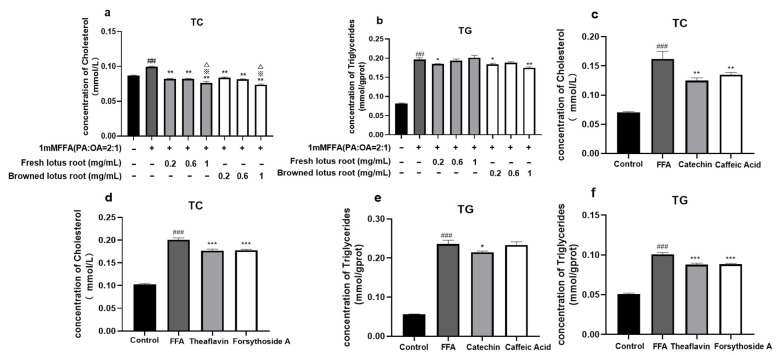
Effects of FLRE (**a**), BLRE (**b**), catechin (**c**), caffeic acid (**d**), theaflavin (**e**), and forsythoside A (**f**) on TC and TG levels in HepG2 cells. HepG2 cells were exposed to 1 mM FFA for 24 h and then treated with FLRE, BLRE, and the monomer compounds for 24 h. Cells treated with solvents were used as the control group, 1 mM FFA was the model group. Significant at ## *p* < 0.01 and ### *p* < 0.001 compared to the control group; Significant at * *p* < 0.05, ** *p* < 0.01 and *** *p* < 0.001 compared to the model group; Significant at ※ *p* < 0.01 compared to the 0.2 mg/mL FLRE group; Significant at △ *p* < 0.01 compared to the 0.6 mg/mL FLRE.

**Figure 3 foods-12-01781-f003:**
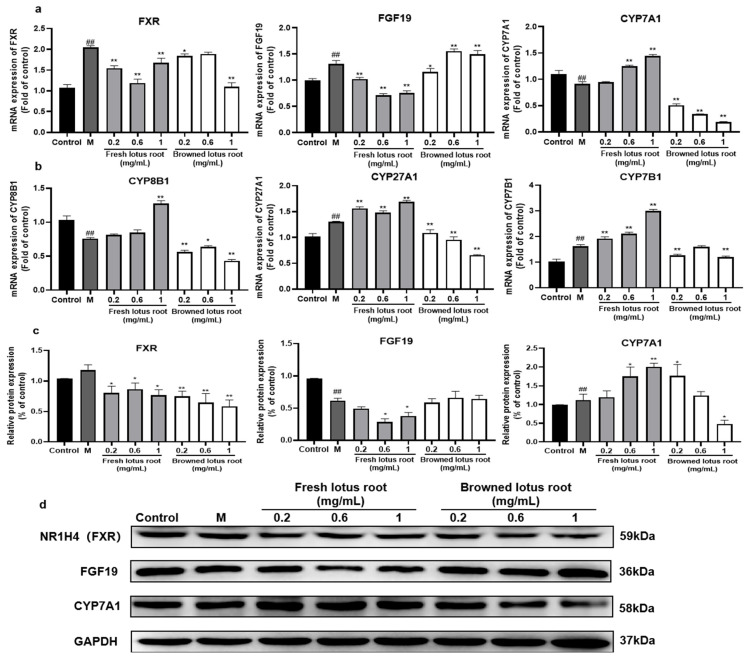
Effects of FLRE and BLRE on the expression of genes and proteins related to cholesterol biotransformation in HepG2 cells induced by FFA. HepG2 cells were exposed to 1 mM FFA for 24 h and then treated with various concentrations of FLRE and BLRE (0.2, 0.6, 1 mg/mL) for 24 h. (**a**,**b**) Total RNA was isolated for RT-PCR analysis of FXR, FGF19, CYP7A1, CYP8B1, CYP27A1, and CYP7B1. (**c**) Total proteins were extracted for Western blotting analysis of FXR, FGF19, and CYP7A1, and quantified by band intensity (**d**). Cells treated with solvents were used as the control group; M was the model group. Significant at ## *p* < 0.01 compared to the control group; Significant at * *p* < 0.05 and ** *p* < 0.01 compared to the model group.

**Figure 4 foods-12-01781-f004:**
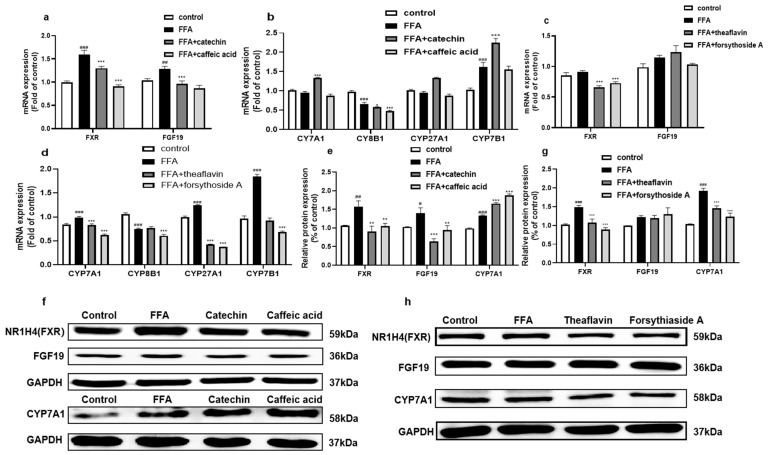
Effects of the monomer compounds (catechin, caffeic acid, theaflavin, and forsythoside A) on the expression of genes and proteins related to cholesterol biotransformation in HepG2 cells induced by FFA. HepG2 cells were exposed to 1 mM FFA for 24 h and then treated with standards for 24 h. (**a**–**d**) Total RNA was isolated for RT-PCR analysis of FXR, FGF19, CYP7A1, CYP8B1, CYP27A1, and CYP7B1. (**f**,**h**) Total proteins were extracted for Western blotting analysis of FXR, FGF19, and CYP7A1, and quantified by band intensity (**e**,**g**). Cells treated with solvents were used as the control group; 1 mM FFA is the model group. Significant at ## *p* < 0.01 and ### *p* < 0.001 compared to the control group; Significant at * *p* < 0.05, ** *p* < 0.01 and *** *p* < 0.001 compared to the model group.

**Figure 5 foods-12-01781-f005:**
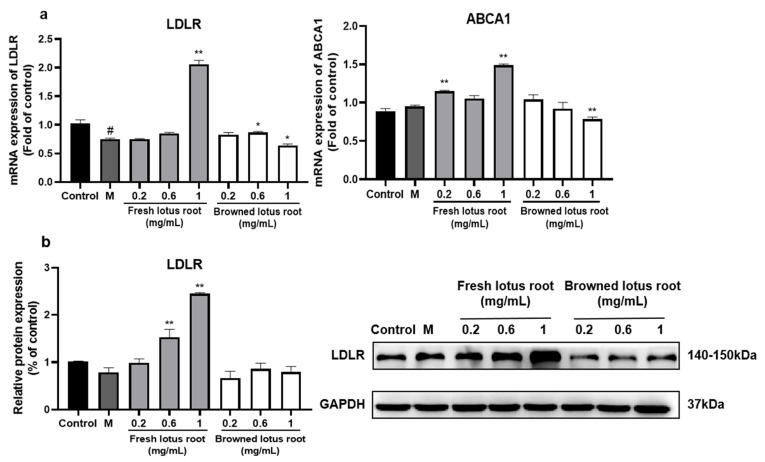
Effects of FLRE and BLRE on the expression of genes and proteins related to cholesterol clearance and efflux in HepG2 cells induced by FFA. HepG2 cells were exposed to 1 mM FFA for 24 h and then treated with various concentrations of FLRE and BLRE (0.2, 0.6, 1 mg/mL). (**a**) Total RNA was isolated for RT-PCR analysis of LDLR and ABCA1. (**b**) Total proteins were extracted for Western blotting analysis of LDLR and quantified by band intensity. Cells treated with solvents were used as the control group; M is the model group. Significant at # *p* < 0.05 compared to the control group; Significant at * *p* < 0.05 and ** *p* < 0.01 compared to the model group.

**Figure 6 foods-12-01781-f006:**
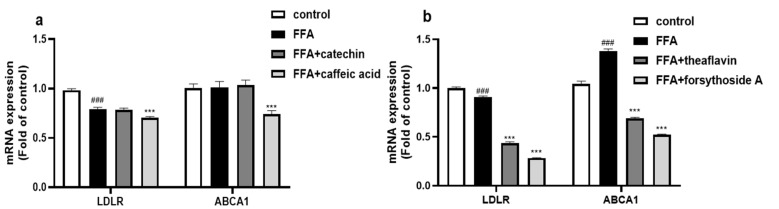
The effects of the monomer compounds (catechin, caffeic acid, theaflavin, and forsythoside A) on the expression of genes and proteins related to cholesterol synthesis in HepG2 cells induced by FFA. HepG2 cells were exposed to 1 mM FFA for 24 h and then treated with the monomer compounds for 24 h. (**a**,**b**) Total RNA was isolated for RT-PCR analysis of SREBP2 and HMGCR. Cells treated with solvents were used as the control group; 1 mM FFA was the model group. Significant at ### *p* < 0.001 compared to the control group; Significant at *** *p* < 0.001 compared to the model group.

**Figure 7 foods-12-01781-f007:**
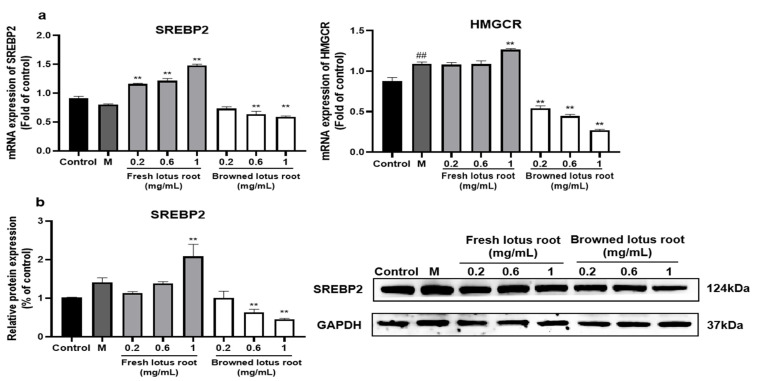
Effects of FLRE and BLRE on the expression of genes and proteins related to cholesterol synthesis in HepG2 cells induced by FFA. HepG2 cells were exposed to 1 mM FFA for 24 h and then treated with various concentrations of FLRE and BLRE (0.2, 0.6, 1 mg/mL) for 24 h. (**a**) Total RNA was isolated for RT-PCR analysis of SREBP2 and HMGCR. (**b**) Total proteins were extracted for Western blotting analysis of SREBP2 and quantified by band intensity. Cells treated with solvents were used as the control group; M is the model group. Significant at ## *p* < 0.01 compared to the control group. Significant at ** *p* < 0.01 compared to the model group.

**Figure 8 foods-12-01781-f008:**
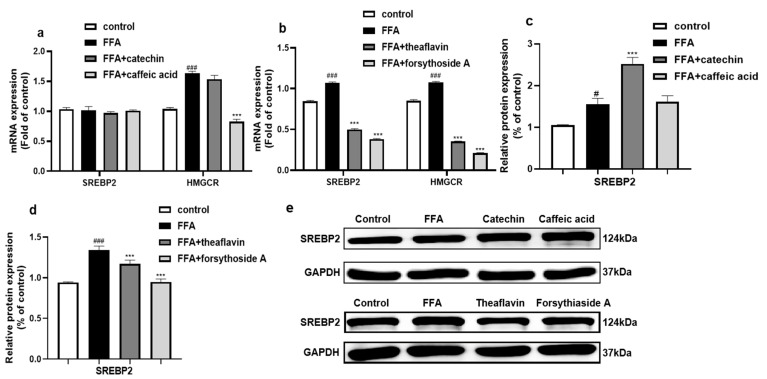
Effects of the monomer compounds (catechin, caffeic acid, theaflavin, and forsythoside A) on the expression of genes and proteins related to cholesterol synthesis in HepG2 cells induced by FFA. HepG2 cells were exposed to 1 mM FFA for 24 h and then treated with the monomer compounds for 24 h. (**a**,**b**) Total RNA was isolated for RT-PCR analysis of SREBP2 and HMGCR. (**c**–**e**) Total proteins were extracted for Western blotting analysis of SREBP2 and quantified by band intensity. Cells treated with solvents were used as the control group; 1 mM FFA was the model group. Significant at # *p* < 0.05 and ### *p* < 0.001 compared to the control group; Significant at *** *p* < 0.001 compared to the model group.

**Table 1 foods-12-01781-t001:** Primers for quantitative RT-PCR.

Gene	Foward (5′-3′)	Reverse (5′-3′)
GAPDH	TGCACCCACCAACTGCTTAGC	GGCATGGACTGTGGTCATGAG
FXR	ACAGAACAAGTGGCAGGTC	CTGAAGAAACCTTTACACCCCTC
FGF19	CTGGAGATCAAGGCAGTCGC	TGCTTCTCGGATCGGTACAC
CYP7A1	GCTTGAGGCACGAGAACCT	GAAAGTCGCTGGAATGGTGT
CYP8B1	CCCTCTCCTTTGGCTCCATCCTC	GCTTGGTGCTGGCTGAGTGTATC
CYP27A1	ACTGCACCAGTTACAGGTGCTTTACA	CCATGTCGTTCCGTACTGGGTACT
CYP7B1	CAGCAGTGCGTGACGAAATTGAC	TGTTCTCTGGTGAGGTGGATGGG
SREBP2	CTCACCTTCCTGTGCCTCTC	AGGCATCATCCAGTCAAACC
HMGCR	TCGCCGACAGTTACTTTCCAAGAAG	TCACAACAAGCTCCCATCACCAAG
LDLR	AGTTGGCTGCGTTAATGTGACA	TCTCTAGCCATGTTGCAGACTTTG
ABCA1	GACATCGTGGCGTTTTTGG	CGAGATATGGTCCGGATTGC

## Data Availability

The data presented in this study are available on request from the corresponding authors.

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
