# Peer review of "Fresh and Browned Lotus Root Extracts Promote Cholesterol Metabolism in FFA-Induced HepG2 Cells through Different Pathways"

_foods, 2023, doi:10.3390/foods12091781_

Round 1
Reviewer 1 Report
15 in vitro, / Italic Consider in the entire document.
16 FFA (specify acronyms)
44 Suggest using capital letters Fibroblast growth factor 19 (Fibroblast Growth Factor)
114 The fractions of the fresh lotus root extracts (FLRE) and browned lotus root extracts (BLRE) were stored at -20°C. The meaning of the acronyms has already been mentioned
118- High sugar medium (DMEM) Dulbecco's Modified Eagle Medium (DMEM)
121 suggest using the word confluence
131 cells without capital letter
136 suggest using the word confluence
159 unclear was conFig.d: 4.4 ???
175 punctuations
266: The word ¨These¨ should be written in lower case.
267 and 269: The next word after the semicolon (;) must be lower case. In Figure 3, the letter C of section is larger than the rest of the letters. It is suggested to change the size for homogeneity in the text.
285 punctuations
317 and 346: The sentence ¨cells treated…¨ must begin with a capital letter.
316 Capital in Western Blot
333: The following wording is suggested: reduced FXR mRNA expression instead of reduced the expression of FXR mRNA expression.
Line 383: The sentence beginning A recent study found, is repeated on line 385.
435 punctuations
In general homogenize Size, improve the quality of figures, the letters are very small and are not clear in the graphics
An interesting well structured and supported research, the conclusions are of great importance to understand the effect in a cellular assay
Author Response
请参阅附件。

Reviewer 2 Report
The manuscript is nicely written, concise and explains all the necessary steps in the analysis and all of the obtained results. However, there are some minor style errors – namely, names of plant species and certain phrases (e.g. in vivo, in vitro) should be italicized throughout the manuscript.
Reviewer 3 Report
The topic of the research article is of great interest. However, it requires lots of improvement. The main drawbacks of this manuscript
Below are several specific comments.
1. The English writing should be further improved, as there are some grammatical or typing errors. It is suggested to ask a native speaker to polish it.
2. Abbreviations should be written on the first appear
3. The abstract should be revised according to the journal's instruction.
Reviewer 4 Report
The study aims to compare the cholesterol metabolism mechanism of fresh and browned lotus root extracts and evaluate the impact of their main component monomers on cholesterol metabolism. The research was well-planned, and it provides interesting insights into cholesterol metabolism. The methods used in this study are appropriate, and the results are thoroughly discussed and explained. However, some concerns should be considered to improve the understanding of this study.
- the abbreviations used such as FFA, TC, and TC should be defined the first time they are mentioned in the text;
- the sub-subsection titles (3.4.1, 3.4.2, etc.) are conclusions of their respective sections and should be given appropriate titles;
- Line 295, "ULRE" should be corrected to its correct term.
Author Response
请参阅附件。
